# Transcriptomic Evaluation of Pulmonary Fibrosis-Related Genes: Utilization of Transgenic Mice with Modifying p38 Signal in the Lungs

**DOI:** 10.3390/ijms21186746

**Published:** 2020-09-14

**Authors:** Shuichi Matsuda, Jun-Dal Kim, Fumihiro Sugiyama, Yuji Matsuo, Junji Ishida, Kazuya Murata, Kanako Nakamura, Kana Namiki, Tatsuhiko Sudo, Tomoyuki Kuwaki, Masahiko Hatano, Koichiro Tatsumi, Akiyoshi Fukamizu, Yoshitoshi Kasuya

**Affiliations:** 1Department of Biomedical Science, Graduate School of Medicine, Chiba University, Chiba City, Chiba 260-8670, Japan; elliptical-full.shuichi@nifty.com (S.M.); hatanom@faculty.chiba-u.jp (M.H.); 2Department of Respirology, Graduate School of Medicine, Chiba University, Chiba City, Chiba 260-8670, Japan; myfortune@nifty.com (Y.M.); tatsumi@faculty.chiba-u.jp (K.T.); 3Life Science Center for Survival Dynamics, Tsukuba Advanced Research Alliance (TARA), University of Tsukuba, Tsukuba, Ibaraki 305-8577, Japan; kimjd75@tara.tsukuba.ac.jp (J.-D.K.); jishida@tara.tsukuba.ac.jp (J.I.); kmurata@md.tsukuba.ac.jp (K.M.); akif@tara.tsukuba.ac.jp (A.F.); 4Laboratory Animal Resource Center and Trans-Border Medical Research Center, University of Tsukuba, Tsukuba, Ibaraki 305-8575, Japan; bunbun@md.tsukuba.ac.jp; 5Graduate School of Sciences and Technology, University of Tsukuba, Tsukuba, Ibaraki 305-8572, Japan; s2021036@u.tsukuba.ac.jp; 6Department of Biochemistry and Molecular Pharmacology, Graduate School of Medicine, Chiba University, Chiba City, Chiba 260-8670, Japan; cada0191@chiba-u.jp; 7Chemical Biology Core Facility and Antibiotics Laboratory, RIKEN Advanced Science Institute, Wako, Saitama 351-0198, Japan; sudo@riken.jp; 8Department of Physiology, Graduate School of Medical and Dental Sciences, Kagoshima University, Kagoshima City, Kagoshima 890-8544, Japan; kuwaki@m3.kufm.kagoshima-u.ac.jp

**Keywords:** p38 mitogen-activated protein kinase, bleomycin-induced pulmonary fibrosis, idiopathic pulmonary fibrosis, RNA sequencing, alveolar epithelial type II cells

## Abstract

Idiopathic pulmonary fibrosis (IPF) is a progressive fibrosing lung disease that is caused by the dysregulation of alveolar epithelial type II cells (AEC II). The mechanisms involved in the progression of IPF remain incompletely understood, although the immune response accompanied by p38 mitogen-activated protein kinase (MAPK) activation may contribute to some of them. This study aimed to examine the association of p38 activity in the lungs with bleomycin (BLM)-induced pulmonary fibrosis and its transcriptomic profiling. Accordingly, we evaluated BLM-induced pulmonary fibrosis during an active fibrosis phase in three genotypes of mice carrying stepwise variations in intrinsic p38 activity in the AEC II and performed RNA sequencing of their lungs. Stepwise elevation of p38 signaling in the lungs of the three genotypes was correlated with increased severity of BLM-induced pulmonary fibrosis exhibiting reduced static compliance and higher collagen content. Transcriptome analysis of these lung samples also showed that the enhanced p38 signaling in the lungs was associated with increased transcription of the genes driving the p38 MAPK pathway and differentially expressed genes elicited by BLM, including those related to fibrosis as well as the immune system. Our findings underscore the significance of p38 MAPK in the progression of pulmonary fibrosis.

## 1. Introduction

Pulmonary fibrosis is the result of the end-stage pathological development of existing lung diseases caused by infection, autoimmunity, chronic inflammation, and idiopathy. Idiopathic pulmonary fibrosis (IPF), one of the most common causes of interstitial pneumonia, is characterized by progressive and irreversible fibrotic scar formation in the gas exchange regions of the lung, resulting in organ malfunction. IPF is a devastating lung disease as patients show poor prognosis, with a median survival of 2–5 years as well as increased risks of pulmonary hypertension and lung cancer [1]. A chronic inflammatory process of the lung has long been considered a main potential mechanism underlying IPF [2]. Moreover, innate and adaptive inflammation may contribute to determining the rate of disease progression in patients with IPF [3]. However, the mortality of patients with IPF is correlated with the extent of fibrotic focus formation, which results from the abnormal and excessive accumulation of extracellular matrix (ECM) components, including collagen, fibronectin, and elastin [4]. Hence, recent studies focusing on the behaviors of ECM-producing myofibroblasts in pulmonary fibrosis may also inform the identification of therapeutic options for IPF [5,6,7,8]. In terms of current pharmacological therapies for IPF, while nintedanib and pirfenidone have been approved by the Food and Drug Administration, neither can improve the survival of patients with IPF [9]. Indeed, new beneficial strategies that enable patients with IPF to survive longer and with improved quality of life have been long-awaited.

Among mitogen-activated protein kinases (MAPKs), members of the p38 MAPK family are activated in response to environmental stresses such as inflammatory stimuli by cytokines and Toll-like receptor ligands, osmolality shock, ultraviolet irradiation, oxidative stress, chemotherapeutic drugs, etc. Of the four isoforms (α, β2, γ, and δ) of p38, p38α is ubiquitously expressed in adult tissues and its physiological and pathological roles have been well investigated [10]. p38 MAPKs are activated by dual phosphorylation of the TGY motif within their activation loop by two upstream MAPK kinases (MAP2Ks)— mitogen-activated protein kinase kinase (MKK)-3 and MKK6—that are activated by various types of MAPKK kinases (MAP3Ks) [11]. In addition to this canonical activation pathway, specific binding of transforming growth factor (TGF)-β-activated kinase 1-binding protein 1 to p38α leads to p38α autophosphorylation and activation [12]. TGF-β signaling is one of the most crucial factors in the murine pulmonary fibrosis model and may be potentiated in the pathogenesis of IPF [13,14]. These findings strongly suggest the involvement of p38 signaling in the development of pulmonary fibrosis. In fact, several studies have reported that p38 inhibitors, SB239063 and FR-167653, can ameliorate bleomycin (BLM)-induced pulmonary fibrosis [15,16]. Lipopolysaccharide-induced epithelial-mesenchymal transition (EMT), in the early pulmonary fibrosis process, may be associated with p38 and TGF-β/smad3 signaling pathways [17]. Additionally, macrophage-specific loss of function of forkhead box M1, which inhibits the p38 signaling pathway, exacerbates BLM-induced pulmonary fibrosis [18]. Furthermore, pirfenidone was originally recognized as a small molecule p38γ inhibitor that blocks the synthesis of TGF-β [19]. Hence, the involvement of p38 signaling in the pathogenesis of pulmonary fibrosis is indubitable.

Here, we designed the study to elucidate new therapeutic target genes for IPF based on the notion that p38 positively regulates the development of pulmonary fibrosis. Mice with stepwise changes in the intrinsic activity of p38, specifically in alveolar epithelial type II cells (AEC II), were subjected to the pulmonary fibrosis model by BLM because AEC II could play a critical role in the progression of IPF [20]. RNA sequencing of total RNA derived from the lungs followed by transcriptome analysis was performed.

## 2. Results

### 2.1. Aggravation of BLM-Induced Murine Pulmonary Fibrosis Correlated with Increased Intrinsic p38 Activity in the Lungs

Histopathological assessment revealed that worsening severity of pulmonary fibrosis was associated with increased intrinsic p38 activity in the lungs (Figure 1A,B). At 8 days post-instillation (dpi) of BLM, tissue infiltration of inflammatory cells and thickening of the alveolar interstitium involved in aberrant collagen accumulation were observed. Moreover, these changes proceeded to diffuse and multifocal distributions at 15 dpi. These histopathological findings of BLM-induced pulmonary fibrosis in the MKK6-constitutive active (MKK6-CA) group were more severe and extensive than those in the wild-type (WT) group, whereas those in the p38-dominant negative (p38-DN) group were less severe and extensive than those in the WT group. Moreover, the distinct severity of pulmonary fibrosis was evident in semi-quantitative evaluation assessed by a modified Ashcroft score and stratified by three mouse groups. In contrast, no apparent inflammatory and fibrotic changes were observed in phosphate-buffered saline (PBS)-treated groups.

In addition, total cell counts in bronchoalveolar lavage fluid (BALF) at 8 dpi of BLM tended to increase with increased intrinsic p38 activity in the lungs (Figure 1C). Regardless of mouse genotype, macrophages and lymphocytes accounted for approximately 50% and 30% of the total cells in the BALF of BLM-treated groups, respectively (Appendix A). Similarly, comprehensive protein analysis in BALF by western blot array revealed 77 upregulated molecules that were evoked by BLM and were associated with increased p38 activity in the lungs (Appendix A). These upregulated molecules included many pro-inflammatory and pro-fibrotic mediators such as interleukin (IL)-13, IL-17, stromal cell-derived factor 1(SDF-1)/C-X-C motif chemokine ligand (CXCL)-12, interferon (IFN)-γ, keratinocyte chemoattractant (KC)/CXCL1, monokine induced by gamma interferon (MIG)/CXCL9, macrophage inflammatory protein-1α (MIP-1α)/CC chemokine ligand (CCL)-3, and regulated upon activation normal T cell expressed and secreted (RANTES)/CCL5.

Measurements of collagen and static compliance in the lungs also supported the morphological alterations in the three mouse groups treated with BLM (Figure 1D,E). The amount of collagen in the left lungs at 8 dpi in the MKK6-CA group was significantly higher than that in the WT and p38-DN groups, although the difference between the WT and p38-DN groups was not significant. The decrease in static compliance in the MKK6-CA and WT groups was significantly larger than that in the p38-DN group, although the difference was not significant between the MKK6-CA and WT groups. Additionally, a reduction in body weight tended to increase with increased p38 signaling in the lungs, exhibiting the systemic effect implicated in the severity of BLM-treated mice (Figure 1F).

We verified the differences in intrinsic p38 activity in the lungs that underlie the different severity of BLM-induced fibrosis among three mouse genotypes. Immunofluorescence staining revealed the presence of AEC II, macrophages, and other parenchymal cells expressing p38 in the PBS-treated lungs (Appendix A). The differences in p38 expression in these lung cells were not observed among three mouse groups. In contrast, the proportion of the lung cells showing the nuclear localization of phospho-p38 (P-p38) was increased by BLM treatment (Appendix A). Additionally, the increased proportion corresponded with the theoretical stepwise upregulation of intrinsic p38 activity in the lungs, and this finding was most prominent in AEC II. Although p38 is ubiquitously expressed in the cytoplasm of resting cells, activated p38 is represented by the phosphorylation-dependent nuclear localization of p38 in response to various types of stimulation such as DNA damage [21,22]. Therefore, these results demonstrate the three graded intensities of p38 activation induced by BLM among three different mouse genotypes.

### 2.2. Comparative Transcriptome Analysis of a BLM-Induced Pulmonary Fibrosis Model Exhibiting Different Severity Due to p38 Activity in the Lungs

RNA sequencing (RNA-seq) was performed using lung samples at 8 dpi when the severity of BLM-induced pulmonary fibrosis was apparently different among the three groups and transcriptomic changes in the BLM-induced fibrosis model are more likely to be correlated with the progression of IPF [23,24]. Principal component analysis (PCA) showed a relationship in the expression of genes among the three mouse groups treated with BLM and PBS, while hierarchical clustering analysis visualized using a heatmap highlighted the trend of differentially expressed genes (DEGs) between the BLM- and PBS-treated groups (Figure 2A). In the PCA plot, the BLM-treated groups were all well separated from the PBS-treated groups and the variance in BLM-treated groups was less than that in PBS-treated groups, indicating the assembly of distinct clusters following BLM exposure. Consistent with this observation, hierarchical clustering analysis identified DEGs between the BLM- and PBS-treated groups. Gene set enrichment analysis (GSEA) in the p38 MAPK pathway revealed that genes involved in regulating this pathway were significantly upregulated in the BLM-treated WT and MKK6-CA groups compared to those in the PBS-treated group (false discovery rate [FDR] q value < 0.25) but not in the p38-DN group (Figure 2B). Moreover, volcano plots in the three mouse groups showed that the increased number of DEGs between the BLM- and PBS-treated groups was associated with an increase in p38 signaling in the lungs (Figure 2C and Appendix A). BLM treatment upregulated approximately two-folds more DEGs and downregulated 2.5-folds more DEGs in the MKK6-CA group than those in the p38-DN group.

Next, we performed transcriptome analysis to detect the enriched functions of DEGs driven by BLM treatment. K-means clustering followed by Kyoto Encyclopedia of Genes and Genomes (KEGG) pathway enrichment analysis revealed the enriched pathways of four clusters in DEGs between the BLM- and PBS-treated groups of three mouse genotypes (Figure 3A). These pathways included fibrosis-related pathways such as protein processing in endoplasmic reticulum (ER) (yellow cluster), cytokine–cytokine receptor interaction (purple cluster), and ECM–receptor interaction (purple and green clusters) in addition to pathways related to immune systems such as hematopoietic cell lineage and leukocyte transendothelial migration. In contrast, we identified 493 common DEGs upregulated by BLM among the three mouse groups (Figure 3B and Appendix A). Regarding 493 common upregulated DEGs, enrichment analysis by gene ontology (GO) revealed three ECM-related annotations among the top five enriched molecular functions and all five terms associated with immune systems among the top five enriched biological processes. Additionally, the cytokine–cytokine receptor interaction was the most significantly enriched pathway among the top five KEGG pathways.

### 2.3. Exploration of Novel Potential Genes Contributing to the Progression of Pulmonary Fibrosis

To identify the pathogenetically relevant genes in the progression of IPF, we investigated the correlation of upregulated genes between BLM-induced fibrotic lungs showing the three different severity levels and human IPF lungs (Figure 4). In the BLM-treated groups, K-means clustering analysis identified a cluster of 2722 genes that their mean reads per kilobase of exon per million mapped sequence reads (RPKM) values increased along with stepwise elevation of p38 signaling in the lungs (Appendix A). We verified 137 DEGs that were included in this cluster and upregulated in common with the three BLM-treated mouse groups. Additionally, human RNA-seq data that provided 475 upregulated DEGs in IPF lung tissue compared to healthy lung tissue was obtained from the Gene Expression Omnibus website (accession ID: GSE52463). Finally, comparison of our data with human RNA-seq data identified four overlapping DEGs; namely, EPH receptor A3 (*EPHA3*), POU class 2 homeobox associating factor 1 (*POU2AF1*), SAM domain, SH3 domain and nuclear localization signals 1 (*SAMSN1*), and ectodysplasin A2 receptor (*EDA2R*).

## 3. Discussion

This study investigated the molecular mechanisms involved in the progressive worsening of BLM-induced pulmonary fibrosis in three genotypes of mice carrying stepwise variations of p38 activity in AEC II. We demonstrated that BLM-induced severe inflammation and fibrosis that was correlated with increased p38 activity in the lungs. Transcriptome analysis of this model provided a connection between the progression of pulmonary fibrosis and genes driving ER functions, ECM-cell interaction, and the immune system. Moreover, we identified candidate genes associated with IPF progression in comparison to a publicly available IPF dataset. The results of these comprehensive analyses suggest that the progression of pulmonary fibrosis occurs concurrently with increased p38 activity in AEC II, which provokes the enhancement of inflammation and immune systems. Therefore, this novel model of pulmonary fibrosis serves as a tool for understanding IPF progression.

Addressing the mechanisms contributing to IPF progression can lead to improved prognosis as this complex multi-pathway disease shows heterogeneity in its clinical course [1,4]. The present study applied a severe model of BLM-induced pulmonary fibrosis to study disease progression. Although the BLM-induced pulmonary fibrosis model is insufficient to mimic the pathogenesis of IPF, it has shown high reproducibility and the important mechanisms of pulmonary fibrosis, such as epithelial-mesenchymal crosstalk and TGF-β signaling pathway [25,26]. BLM-induced pulmonary fibrosis shows a transition from the inflammatory to the fibrotic phase at around 7 dpi, establishment of fibrosis at 14 dpi, and subsequent formation of reversible lesions [27,28]. In this context, the analyses were mainly conducted at 8 dpi as the optimal timing to evaluate the progression of pulmonary fibrosis. Additionally, the transcriptome profiling approach enables us to reveal the molecular mechanisms regulating fibrosis in this model and compare them to the profiles in the lung samples of IPF patients. Recent studies have shown that variations in gene expression of BLM-induced pulmonary fibrosis were correlated with changes in IPF severity [23,24]. Notably, the genes differentially expressed in BLM-induced pulmonary fibrosis are most abundant in the active fibrotic phase (7–14 dpi), which shows the highest correlation with IPF lung samples [23,24]. This finding explains the rationale that the gene expression profiles of BLM-treated lungs are altered before remarkable changes in morphology and function occur. Taken together, these findings are compatible with our study, which has implications for the development of pulmonary fibrosis in transcriptome analysis.

Although apoptosis and reprogramming of lung epithelial cells play a prominent role in IPF, the molecular details remain uncertain [20,29]. p38 is required for maintaining AEC II homeostasis as a physiological function, whereas extracellular stimuli-mediated enhancement of p38 is attributed to lung inflammation and immune responses and is associated with apoptosis in AEC II [30,31]. We focused on p38 activity in AEC II to examine pulmonary fibrosis progression and performed transcriptome analysis. The results showed distinct expression of p38 MAPK pathway genes that was positively correlated with stepwise changes in intrinsic p38 activity in the lungs and the contribution of the immune system and ER functions to the development of pulmonary fibrosis mediated by activation of the p38 MAPK pathway. Regarding lung inflammation, the exacerbation correlated with increased p38 activity in the AEC II manifested as increases in inflammatory cells and pro-inflammatory cytokines in BALF and the enrichment of genes facilitating immune cell infiltration and cytokine interaction pathways. Augmentation of pro-fibrotic cytokines and immune response arising from inflammation leads to progression of tissue remodeling and fibrosis in the lungs [32]. In particular, the TGF-β signaling pathway driven by p38 induces EMT and fibroblast proliferation and activation through epithelial–mesenchymal crosstalk [33,34,35]. In this study, TGF-β1 was included in the 137 overlapping genes that showed correlations with intrinsic p38 activity in the lungs and upregulation among the three mouse groups treated with BLM (Figure 4 and Appendix A). Additionally, IL-13 and IL-17, which in BALF were upregulated with a concomitant increase in intrinsic p38 activity in the lungs, can promote TGF-β signaling pathway-dependent EMT and fibroblast proliferation and resistance to apoptosis [36,37]. These cytokines originate from immune cells such as T cells, suggesting an association between the immune system and fibrosis [38,39]. A previous study showed that increased immune cells and aberrant regenerating epithelial cells express inflammatory mediators, including IL-17, in active fibrotic lesions of IPF lungs [40]. Furthermore, single-cell RNA-seq analysis of epithelial cells displaying atypical phenotypes in IPF lungs showed that these epithelial cells modulated the expression of inflammatory response- and TGF-β signaling pathway-related genes, leading to fibrotic remodeling [41]. Collectively, these findings strongly suggest that inflammation and immune response enhanced by increased p38 activity in AEC II may contribute to the fibrotic process in the lungs.

Another possible explanation for the mechanism affecting fibrosis is that maladaptive ER stress response and its mediated apoptosis occurred concomitantly with increased p38 activity in AEC II. ER functions to retain cellular homeostasis by conducting posttranslational modification of proteins, with an adaptive process called unfold protein response under various stress conditions, although an excess of ER stress disrupting this adaptation elicits apoptosis [42]. The enrichment analysis in our study revealed that the ER protein processing pathway activated by BLM-induced reactive oxygen species was correlated with increased p38 activity in the AEC II. This result is consistent with the fundamental principle that ER stress can function in concert with the p38 MAPK pathway [43]. Simultaneously, AEC II homeostasis is sustained by its interaction with ECM, and p38 may participate in it [44]. A recent study using human fibroblasts revealed that the p38 MAPK pathway mediated the acquired resistance of ER stress modified by ECM metabolism through cell-to-ECM interaction [45]. In our study, ECM-receptor interaction was also a pathway enriched in accord with increased p38 activity in AEC II. Moreover, upregulation of ECM-related genes in enrichment analysis and matrix metalloproteases (MMPs; MMP-2, -3, and -9) in BALF (Appendix A), in addition to higher amounts of collagen, were connected to increased p38 activity in AEC II. These results are consistent with the study that MK2, a downstream substrate of p38, engaged in fibroblast activation and ECM production potentiated by fibroblast activation [46]. In addition, MMPs, which degrade all components of the ECM, are regulated by p38, while their upregulation leads to apoptosis and abnormal regeneration of lung epithelial cells [47]. Hence, ER stress in AEC II can be augmented by not only BLM-induced cytotoxicity, but also by the accumulation of ECM and AEC II to ECM interaction, controlled by the p38 MAPK pathway. These results emphasize the importance of p38 activity in AEC II and its related molecules in the progression of pulmonary fibrosis. In contrast, this murine model created by the intratracheal administration of BLM was not followed up after the establishment of pulmonary fibrosis. Therefore, further studies are required to determine whether p38 activity in AEC II influences the restoration of BLM-induced pulmonary fibrosis.

We validated four therapeutic target genes by comparing our data with publicly available data from IPF patients. First, *EPHA3* is expressed predominantly in lymphocytes and encodes a receptor tyrosine kinase implicated in regulating cell adhesion and cellular motility [48]. A recent study demonstrated that a novel epithelial cell population derived from IPF lungs co-expressed EPHA3 and CC chemokine receptor (CCR)-10 and facilitated the development of lung remodeling [49]. CCL28, a chemokine ligand for CCR10, was upregulated in BALF in accordance with increased p38 activity (Appendix A) [50]. These findings suggest that coordination between CCL28-CCR10 chemokine signaling and the p38 MAPK pathway has important implications for reprogramming of epithelial cells, a speculation that warrants further investigation. Second, *POU2AF1* encodes a transcriptional coactivator that regulates B cell maturation and humoral immunity and is expressed in both airway epithelial and B cells [51,52]. A prior study using IPF lungs documented that transcriptome analysis identified *POU2AF1* as a promoter of pulmonary fibrosis and it is highly expressed in aggregates of B cells [53]. Third, *EDA2R* regulates ectodermal tissue development; its expression in lung epithelial cells such as AEC II was ascertained by single-cell RNA-seq data set in IPF lungs [41,54]. Genome-wide association study using human lung tissue identified *EDA2R* as a candidate gene involved in lung aging [55]. In addition, this gene accelerates the apoptotic process in two different types of epithelial cells by activation of p53 signaling and caspase cascade [56,57]. These findings suggest the involvement of *EDA2R* in AEC II senescence and apoptosis. Lastly, *SAMSN1* is expressed in healthy lung epithelial cells but not in lung cancer cells [58]. Although its functions in the lungs remain unknown, this gene is pivotal in regulating B cell activation and differentiation [59]. Thus, changes in the expression levels of these candidate genes by p38 activity may be involved in promoting fibrosis through molecular interactions between epithelial and immune cells in the IPF lung. This hypothesis is supported by two previous reports showing an association of lymphocytes and epithelial cells with progressive fibrosis in transcriptome analysis of IPF lungs [60,61]. Therefore, the interplay between these genes and the p38 MAPK pathway may be key to understanding the immunological mechanisms underlying IPF progression. However, further studies are needed to confirm the clinical significance of these genes in the patients with rapidly progressive IPF.

## 4. Materials and Methods

### 4.1. Mice

All animal procedures conformed to the Japanese regulations for animal care and use, following the Guidelines for Animal Experimentation of the Japanese Association for Laboratory Animal Science. Male and female C57BL/6J mice were purchased from Clea Japan (Tokyo, Japan). Using the 3.7SP-C/SV40 vector kindly provided by Dr. Jeffrey A. Whitsett (Children’s Hospital Medical Center, Division of Pulmonary Biology, Cincinnati, Ohio), we generated the following transgenic mice: C57BL/6J-hSP-C-M2 flag-p38α dominant-negative (d.n., dual mutations in wild mouse p38α: Thr180 to Ala; Tyr182 to Phe) TG (p38-DN) mice [62] and C57BL/6J-hSP-C-3HA-tag-MKK6 constitutive-active (c.a., dual mutations in wild human MKK6: Ser207 to Asp; Tyr211 to Asp) TG (MKK6-CA) mice [63]. We confirmed that each transgene-derived product was expressed at least in surfactant protein C (SP-C)-positive AEC II in the lung by using anti-M2-Flag or anti-HA tag antibody. Male heterozygous TG mice and WT littermates aged 10–12 weeks were used for the experiments. The animals were housed in standard laboratory cages and allowed food and water throughout the experiments. The studies were performed according to a protocol approved by the Committee of Animal Welfare of Chiba University.

### 4.2. BLM-Induced Pulmonary Fibrosis Model

Mice were anesthetized and the neck skin of each was cut longitudinally to expose the trachea. After a single intratracheal instillation of BLM hydrochloride (3 mg/kg; Nippon Kayaku, Tokyo, Japan) dissolved in PBS using a repeating syringe dispenser (Hamilton, Reno, NV, USA), the skin was sutured. Control mice were administered a sham treatment with PBS. Then, changes in body weight were measured daily. To evaluate the histopathological changes in the lung samples at 8 and 15 dpi of BLM, freshly cut lung sections (5 µm thick) were placed on adhesive glass slides (Matsunami Glass Ltd., Osaka, Japan) and stained with Masson’s trichrome. The changes in the fibrotic lung samples were evaluated semi-quantitatively according to the modified Ashcroft method with a scoring grade of 0 to 8 [64]. In addition, the collagen content of the left lung was measured using the Sicol Soluble Collagen Assay Kit (Biocolor Life Science Assays, Carrickfergus, United Kingdom) according to the manufacturer’s protocol.

### 4.3. Evaluation of Inflammatory Cells in BALF

At 8 dpi, the trachea was exposed and lavaged three times with 1 mL ice-cold PBS using a 20-gauge catheter. The BALF was centrifuged at 400× *g* for 10 min and the resulting supernatants were stored at −80 °C for protein array analysis. The resulting cell pellets were resuspended in PBS and subjected to cell counting using a hemocytometer in combination with Diff-Quick (Sysmex Corporation, Kobe, Japan) staining.

### 4.4. Measurement of Left Lung Compliance

As described previously [65], the lung compliance of the mice was measured by drawing static air pressure–volume relationships in a mixture of medetomidine, midazolam, and butorphanol (M/M/B: 0.3/4/5 mg/kg)-anesthetized mice tracheotomized with polyethylene tubing (O.D. = 0.8 mm). Total lung capacity was defined as the lung volume of full inflation judged by visual inspection of the lung that fully occupied the chest cavity. Functional residual volume was defined as deflation at 0 cm H_2_O. Lung volumes at an airway pressure of 20 cm H_2_O were estimated between mice at 8 dpi with BLM and PBS in the three genotypes (WT, p38-DN, and MKK6-CA mice).

### 4.5. Immunofluorescence Staining

The lung sections were pretreated with 1:10 FcR blocking agent (Miltenyi Biotech, Gladbach, Germany) for 10 min. They were then treated with primary antibodies (1:100 dilution) as follows: goat anti-proSP-C polyclonal antibody (sc-7706; Santa Cruz Biotech, Dallas, TX, USA), rabbit anti-p38 polyclonal antibody (original production [66]), rabbit anti-proSP-C antibody (customized production [60]; Sigma-Aldrich Japan Genosys, Ishikari, Japan), and mouse anti-phospho-p38 MAPK (pT180/pY182) (clone30, 612281; BD Biosciences, NJ, USA), followed by staining with appropriate fluorescein-conjugated secondary antibodies (1:200 dilution), and 4′,6-diamidino-2-phenylindole (DAPI) was used for nuclear staining. The stained sections were observed under a fluorescence microscope (Axio Imager A2; Zeiss, Oberkochen, Germany).

### 4.6. RNA Sequencing

At 8 dpi, mice under anesthesia were intracardially perfused with ice-cold PBS to wash out blood cells in the lungs and sacrificed. The left lung lobes were homogenized in ISOGEN plus (TaKaRa Bio, Kusatsu, Japan), and total RNA was extracted. Thereafter, 500 ng of total RNA was ribosomal RNA-depleted using a NEBNext rRNA Depletion Kit (New England Biolabs) and was converted to Illumina sequencing library using NEBNext Ultra Directional RNA Library Prep Kit (New England Biolabs). The library was validated to determine the size distribution and concentration using a Bioanalyzer (Agilent Technologies). Sequencing was performed on a NextSeq 500 (Illumina) instrument with paired-end 36-base read options. Reads were mapped on the mm10 mouse reference genome and quantified using CLC Genomics Workbench version 12.0 (QIAGEN). All RNA-seq data sets were deposited in the Gene Expression Omnibus database at the National Center for Biotechnology Information with accession number GSE154074.

### 4.7. Identification of Differentially Expressed Genes (DEGs)

To estimate the expression patterns of transcripts among the three genotypes (WT, p38α d.n.-TG and MKK6 c.a.-TG mice) with or without BLM instillation, the read counts were normalized by calculating the number of reads per kilobase per million for each transcript in individual samples using CLC Genomics Workbench version 12.0 (QIAGEN) [67]. Filtering characteristics of fold change −2 to 2 (FDR at *p* < 0.05) were used to identify the DEGs. Subsequently, the distinct gene expression patterns were analyzed comparatively through PCA and clustering heatmaps using CLC Genomics Workbench. GSEA for p38 MAPK pathways in the BLM-treated group among the three genotypes was also performed using GSEA_4.0.3. [68]. K-means functional enrichment analysis of DEGs was analyzed using integrated differential expression and pathway analysis (iDEP) online tools [69]. A volcano plot was used to compare the gene expression levels in terms of the log_2_ fold change. The GO (molecular function and biological process) and KEGG pathway analyses of DEGs between BLM- and PBS-treated groups were performed using ToppGene Suite (https://toppgene.cchmc.org) [70]. Finally, K-means cluster analysis was performed to identify BLM-upregulated genes that depended on the theoretical intrinsic activity of the p38 signal (p38-DN < WT < MKK6-CA) using CLC Genomics Workbench.

### 4.8. Statistical Analysis

Data are expressed as means ± standard error of the mean (SEM). Statistical analysis was conducted using GraphPad Prism Version 6 (GraphPad Software, San Diego, CA, USA). Statistical significance was determined by one-way analysis of variance (ANOVA) followed by Tukey’s or Student’s *t*-tests, and *p*-values < 0.05 were considered significant.

## Figures and Tables

**Figure 1 ijms-21-06746-f001:**
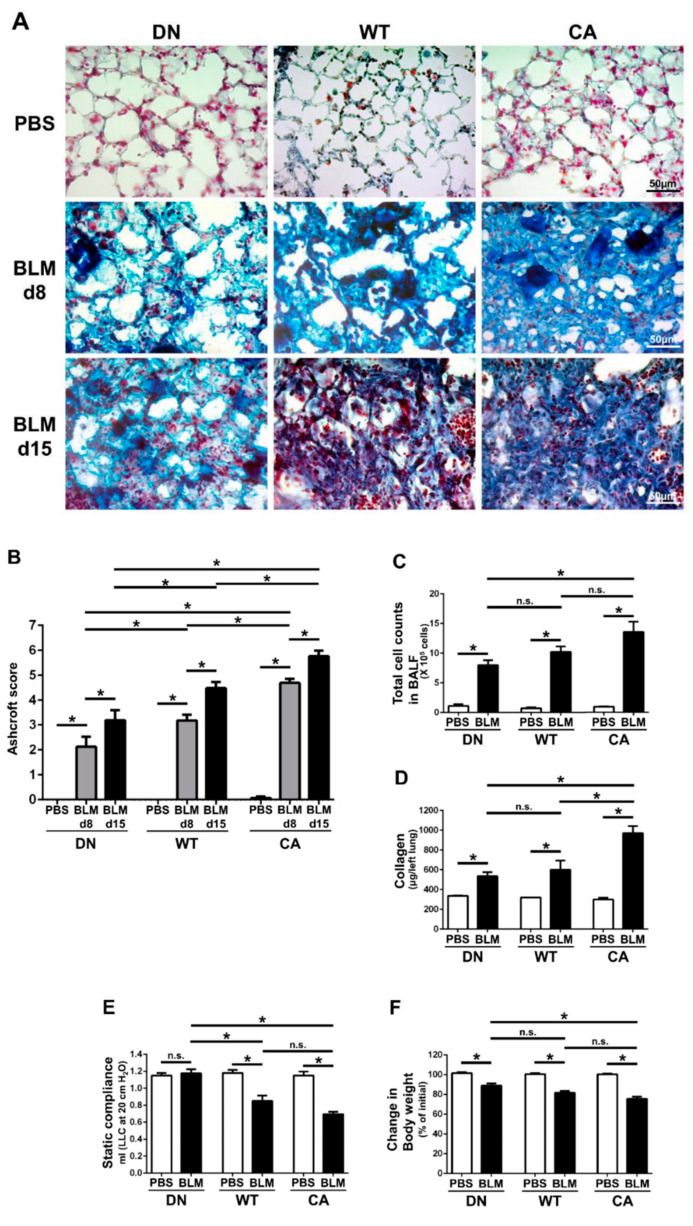
Bleomycin (BLM)-induced pulmonary fibrosis in mice bearing three different abilities of p38 in the lungs. The three mouse genotype groups; namely, MKK6 constitutive active group (CA), wild type group (WT), and p38 dominant negative group (DN), were intratracheally administrated BLM and phosphate-buffered saline (PBS). (**A**) Representative histopathological images of the lung sections stained by Masson’s trichrome (scale bar = 50 µm). Lungs were collected at 8 days post-instillation (dpi) of BLM and PBS, and 15 dpi of BLM. (**B**) Quantification of the fibrotic severity using modified Ashcroft scoring was evaluated in six different lesions at 8 dpi of BLM and PBS, and 15 dpi of BLM (n = 9). (**C**) The numbers of total cells in bronchoalveolar lavage fluid were measured at 8 dpi of BLM and PBS (n = 7). (**D**) The collagen contents of the left lung lobes were measured at 8 dpi of BLM and PBS and normalized to the weight of each left lung (n = 4). (**E**) The static lung compliances were measured at 8 dpi of BLM and PBS (n = 4). (**F**) Proportions of body weight at 8 dpi of BLM and PBS to that before administration (n = 14). All data are represented as means ± standard error of the mean (SEM). * *p* < 0.05, n.s., no significant difference (measured by one-way an analysis of variance (ANOVA) followed by Tukey’s test or unpaired Student’s *t*-test).

**Figure 2 ijms-21-06746-f002:**
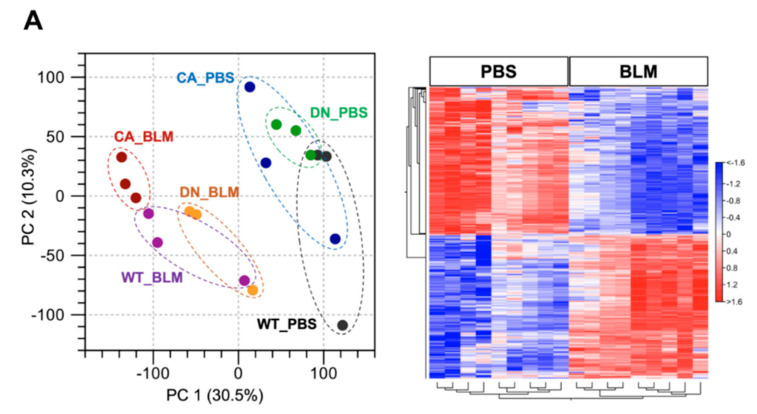
Expression profiling of BLM- and PBS-treated lungs in three different mouse genotypes. Three samples from each group were sequenced. (**A**) Principal component analysis of RNA sequencing datasets among the BLM- and PBS-treated three mouse groups (left). Hierarchical clustering shown in a heatmap of gene expression profiles between the BLM- and PBS-treated groups (right). The red and blue strips represent upregulated and downregulated genes in each group, respectively. (**B**) Gene set enrichment analysis of differential expression in the p38 MAPK pathway between the BLM- and PBS-treated groups of three mouse genotypes. The normalized enrichment scores (NES), normal *p*-values, and false discovery rate (FDR) q values are indicated. (**C**) Volcano plot of differentially expressed genes altered by BLM treatment among three mouse groups. Upregulated and downregulated genes are discriminated based on log2 fold-change and adjusted FDR *p*-value (<0.05).

**Figure 3 ijms-21-06746-f003:**
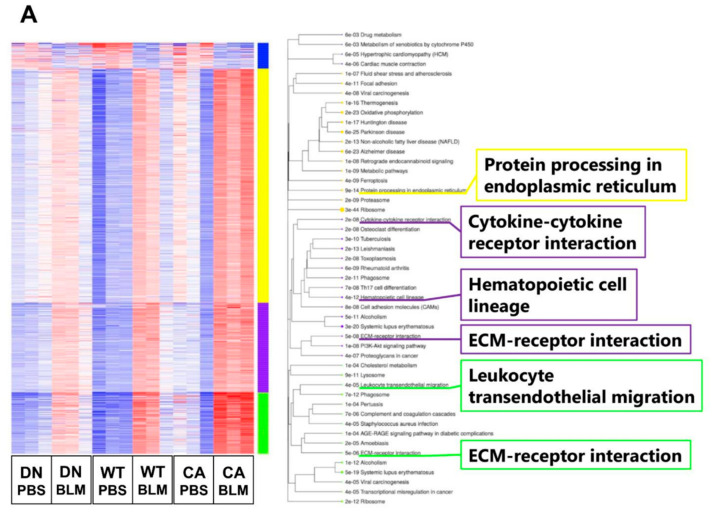
Gene ontology (GO) and Kyoto Encyclopedia of Genes and Genomes (KEGG) pathway enrichment analysis of differentially expressed genes (DEGs) altered by BLM treatment among three mouse groups. (**A**) K-means clustering shown in a heatmap based on the gene expression profiles between the BLM- and PBS-treated groups of three mouse genotypes (left). The colored bars on the right of the diagram indicate clusters. The trees represent enriched KEGG pathways corresponding to each cluster (right). (**B**) Venn diagram showing the overlap of DEGs upregulated by BLM among three mouse groups, with the numbers of DEGs indicated in each area (left). GO and KEGG pathway enrichment analysis of 493 common upregulated DEGs among the three genotypes (right). Top five enriched GO terms associated with molecular function (upper) and biological process (middle), and KEGG pathway analysis (bottom).

**Figure 4 ijms-21-06746-f004:**
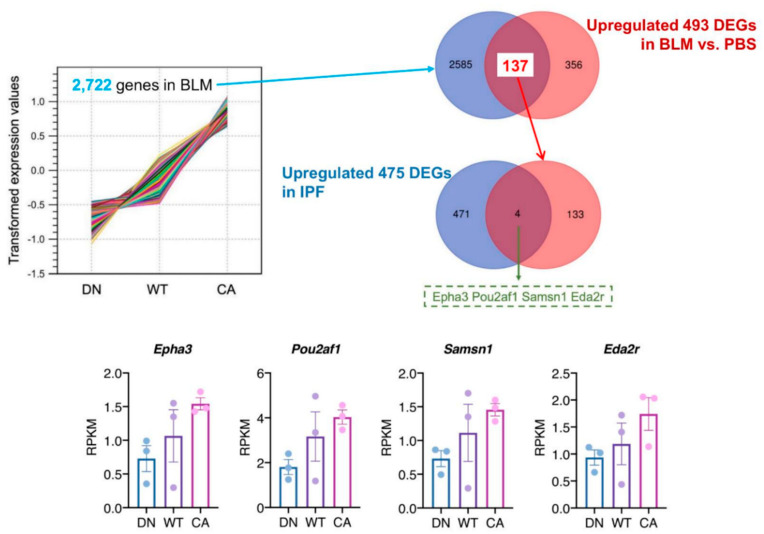
Identification of potential target genes by comparison to a publicly available idiopathic pulmonary fibrosis (IPF) dataset. K-means cluster analysis among three mouse groups treated with BLM revealed a cluster of 2722 genes showing correlation between variations of their mean expression values and stepwise changes in intrinsic p38 activity in the lungs (**left upper**). The Venn diagram in the right upper tier shows the overlap of 2722 genes in this cluster and 493 common upregulated DEGs among the three mouse groups. Likewise, the Venn diagram in the right middle tier shows the overlap of 137 genes identified in our study and 475 upregulated DEGs in human IPF lungs from dataset GSE52463. The four overlapping genes identified in these analyses were EPH receptor A3 (*EPHA3*), POU class 2 homeobox associating factor 1 (*POU2AF1*), SAM domain, SH3 domain and nuclear localization signals 1 (*SAMSN1*), and ectodysplasin A2 receptor (*EDA2R*) (**bottom**). Each bar and plot represent mean reads per kilobase of exon per million mapped sequence reads (RPKM) ± SEM and RPKM value of each sample, respectively.

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
