# Peer review of "Transcriptomic Evaluation of Pulmonary Fibrosis-Related Genes: Utilization of Transgenic Mice with Modifying p38 Signal in the Lungs"

_ijms, 2020, doi:10.3390/ijms21186746_

Round 1

Reviewer 1 Report

The authors of the manuscript entitled “Transcriptomic evaluation of pulmonary fibrosis-related genes: Utilization of transgenic mice with modifying p38 signal in the lungs” provide a strong link between the increased severity of BLM-induced pulmonary fibrosis and elevated activity of p38 signaling in the lungs. For this reason they used a robust model: transgenic mice carrying stepwise variations in intrinsic p38 activity in the AEC II. The results of transcriptome analyzis are rather interesting and could shed new light on the understanding of molecular mechanisms underlying progression of pulmonary fibrosis. The Discussion section provides good explanation of the results and makes them more valuable.

The article is well written, addresses all the questions raised and should be interesting for the specialists in the field. It could be recommended for publication in the present form.

Author Response

Thank you for positive comments and careful review on our paper.

Reviewer 2 Report

Title: Transcriptomic evaluation of pulmonary fibrosis-related genes: Utilization of transgenic mice with modifying p38 signal in the lungs This is a descriptive manuscript by Matsuda et al characterized transcriptomic changes in the lungs of “epithelial p-38 transgenic mice” following bleomycin-induced lung fibrosis. The authors established MAPK-p38 transgenic mice to elucidate the significance of the p38 MAPK pathway in lung fibrosis pathogenesis. Although the significance of p38MAPK is known in lung fibrosis, additional information would still be useful. However, bulk RNAseq generally now provides limited utility especially when functional assay to support the hypothesis is not available. And therefore, the results may not as comprehensive and informative as the authors had claimed. Few issues need to be addressed before meaningful conclusions can be reached. 1. The data derived from transgenic mice that require further characterization; first, p-38 MAPK expression on AECII needs to be demonstrated either by immunostaining or flow cytometry. Also, please further clarify that P38-MAPK expression of other cell types is not affected in these mice. In the PCA plot, even though CA mice seem to be well-clustered, there is some overlap between WT and DN.  2. In addition to transcriptomic data, please include data of p38-MAPK activity (by WB or immunostaining)  in the lungs at the normal and bleomycin-injured stated, or providing a clear reference if these data were previously published.  3. Further characterization of BAL cells beyond cell counts; at least differential or flow cytometry to identify subtypes of cells in BAL. 4. Please provide qPCR to validate the significance of all 4 genes (Epha3, Pou2af1, Samsn1, Eda2r) in figure 4. 

Round 2

Reviewer 2 Report

I appreciate some improvement in the revised manuscript and believe that the authors have adequately addressed the concerns.

A few minor comments;

1. Supplemental figure 2A-B. Although the quantitation (Suppl Fig 2B) may not reflect the findings in the pictures (Suppl Fig 2A) because only specific fields of the staining can be depicted, I have some difficulty to appreciate that the activity of P-p38 nuclear localization is significantly higher at baseline or after bleomycin injury between WT and CA lungs (Supp Fig 2A). If able, I would select a better picture to represent the quantitative data (Supp Fig 2B).

2. Also for immunofluorescent staining, if space permits, it would be best to also show a single channel of each immuno-staining.

Author Response

Response to Reviewer 2:

  1. Supplemental figure 2A-B. Although the quantitation (Suppl Fig 2B) may not reflect the findings in the pictures (Suppl Fig 2A) because only specific fields of the staining can be depicted, I have some difficulty to appreciate that the activity of P-p38 nuclear localization is significantly higher at baseline or after bleomycin injury between WT and CA lungs (Supp Fig 2A). If able, I would select a better picture to represent the quantitative data (Supp Fig 2B).
  2. Also for immunofluorescent staining, if space permits, it would be best to also show a single channel of each immuno-staining.

Response for 1 and 2: We appreciate your helpful suggestions. We have incorporated these suggestions into the revised version as much as possible. We have divided Supplementary Figure 2A into Supplementary Figure 2A and 2B, and have changed Supplementary Figure 2B to Supplementary Figure 2C. The corrected parts in the Supplementary Figure 2 are highlighted using the "Track Changes" function. We also have replaced “Supplementary Figure 2A and 2B” with “Supplementary Figure 2B and 2C” in the Results section (manuscript, page 3, line 132-133).

In Supplementary Figure 2B, we have added representative images of immunofluorescence staining for P-p38 and proSP-C of the PBS-treated lungs. The presence of AEC II with P-p38+ nuclei was evident in the CA lung, although few of them exist in the WT and DN lungs. Additionally, we have revised the images of the BLM-treated lungs and have added those of staining for P-p38 and DAPI, and proSP-C. These images explained the differences in the number of P-p38 nuclear localization especially in AEC II among the three genotypes.

Thank you once again for your valuable comments. We hope these revisions meet your expectations.